# Factors Affecting Autistic Students' School Motivation

Chandra Lebenhagen [1,*] and Jaclyn Dynia [2]

1   Werklund School of Education, University of Calgary, Calgary, AB T2N 1N4, Canada
2   SproutFive's Center for Early Childhood Innovation, Columbus, OH 43212, USA; jdynia@sproutfive.org
*   Correspondence: chandra.lebenhagen@ucalgary.ca

**Abstract:** Very little identifiable research explores the factors impacting autistic students' school motivation and how these factors may or may not affect their academic and well-being outcomes in secondary school and beyond, including post-secondary enrollment, employment, and their quality of life. Instead, research on autism and inclusive education mainly focuses on the efficacy of interventions aimed at teaching skills related to sensory, communication, social, and behavior. **Methods:** A secondary analysis of survey data from an original mixed-method study was conducted to investigate how environmental, teacher, and peer factors are associated with autistic students' school motivation. Eligible participants were between the ages of 15 and 21. In total, 72 participants (*n* = 72) completed an online survey to share their perspectives on their school experiences. Subsequently, linear regression analysis was conducted to answer the research question. **Results:** Participants who rated their schools as having pleasant physical spaces and felt that their peers understood them as a person had higher levels of school motivation. Teachers were not found to be significantly related to students' school motivation. Participants who shared that typing was their preferred mode of communication were less motivated to attend school than students who preferred speaking communication. **Conclusions:** Environmental and peer factors are related to more than the day-to-day school experiences of autistic students; they are also related to their school motivation. These findings add to the existing literature on inclusive education and positive school outcomes for autistic students and offer additional explanations of the barriers that affect autistic students' graduation from secondary school and interest in attending post-secondary education.

**Keywords:** autism; autistic voice; school motivation; inclusive education; secondary; post-secondary

## 1. Introduction

There has been a recent shift in the field of autism research to focus on autistic adults' experiences and views [1] and, most notably, how to improve lifelong outcomes for autistic individuals [2]. A critical outcome for autistic adults is educational attainment [3,4]. Unfortunately, educational attainment for autistic adults is lower than for the general population [5,6]. For example, research finds that, despite having a strong desire to continue their education after secondary school [7], approximately 44% of autistic students in the United States enroll in post-secondary education, compared to 70% of the general population [5,6]. Unfortunately, comparative data for Canadian post-secondary autistic students is unavailable [8]. However, research finds that of Canada's 258 publicly funded post-secondary institutions, 15 offered at least one autism-specific support program, with 67% being website information and 47% being transition support [8]. Academic and non-academic factors are related to autistic students' post-secondary success [9,10]. For instance, post-secondary students report that exam stress [11], inadequate access to individualized support [9], and difficulties with planning, organization, and time management [12] negatively impact their academic success. Non-academic barriers reported by autistic post-secondary students include navigating the sensory aspects of large class sizes and crowded campuses [13], feeling socially isolated and lonely [14], and bullying [15]. Moreover, these academic and non-academic factors prevent students from completing their post-secondary programs [16].

Research shows that many of these barriers may also be present in secondary school. For instance, when asked to report on their secondary school experiences, autistic students share expectations of handwriting [17], test anxiety [18], the pace of instruction, and the availability of support [13,19,20], which impact their school achievements, along with non-academic factors, such as sensory stimuli [13], social–communication breakdowns [9], bullying, unpredictability, and feelings of anxiety [18]. Repeated exposure to undesirable academic and non-academic factors results in disproportionate levels of non-attendance for autistic secondary school students [21].

While similarities exist between the academic and non-academic barriers to school success for autistic students in secondary and post-secondary settings, little research has investigated the associations between these factors and school motivation. Motivation often refers to the factors affecting students' attitudes, interest, and engagement in learning [22], which are influenced by internal and external factors such as the psychological states affecting mood and how stimulating and accessible a learning task is [23]. Motivation is a fundamental aspect of achievement [24] because it promotes higher-level cognition, creativity, and perseverance [25]. Of the available research on factors affecting autistic students' school motivation, most focus on the perspectives of students enrolled in post-secondary, which finds that autistic students who experience difficulties with communication, social interactions, and overwhelming school environments are less motivated to enroll and complete their post-secondary school programs [20]. Alternatively, factors found to reduce post-secondary dropouts and improve graduation rates include an instructor's knowledge of autism, peer understanding and acceptance, family support, and post-school goals [9,20,26].

While no identifiable research exists on the factors affecting school motivation for secondary school students, autistic students report that school environments, relationships with peers and teachers, and access to reliable communication impact their sense of school belonging and participation [13,27,28]. Autistic secondary school students also report that when their teachers have more than a basic understanding of autism and take the time to get to know them in personally meaningful ways, they feel more understood, and the support they receive is more effective [19]. Subsequently, since similarities exist between the academic and non-academic factors that may influence the school experiences of autistic students, the purpose of this study was to complete a secondary analysis of perception data from autistic secondary school students to investigate the factors affecting their school motivation. Based on previous research on the factors affecting the school motivation of post-secondary students, the authors hypothesized that difficulties with communication, social interactions, and the sensory aspects of school environments may also be involved in the school motivation of autistic students attending secondary school. Conducting a deeper exploration of the academic and non-academic factors impacting autistic students' school motivation is beneficial to reduce and remove the attitudinal and systemic barriers preventing graduation from secondary school and enrollment rates in post-secondary programs for autistic students.

## 2. Materials and Methods

This study conducted a secondary analysis of the survey data from an original mixed-method study that investigated the self-reported school experiences of secondary school autistic students [28]. Qualitative data from the open-ended survey question was not integrated into the current analysis because most of the 19 responses centered on students either sharing their like or dislike for school, such as "Yes, My School Life is Wonderful. Enjoying [it] Very Much" or "School was a horrible experience". However, qualitative findings from the semi-structured interviews (10 participants) are briefly presented in Section 3 of this paper and strengthen the validity of the current findings.

*2.1. Participants*

Participants were recruited using a convenience sampling method, specifically snowball sampling [29]. Recruitment emails were sent to publicly available contacts at provincial and territory autism stakeholder groups and through Amazon's Mechanical Turk (MTurk).

According to Canada's Tri-Council Policy Statement: Ethical Conduct for Research Involving Humans (TCPS 2, 2018), participants aged 14–17 qualify as mature minors and can be provided with an adult/participant consent form if they possess decision-making capacity. Aligning with these federal recommendations and guidelines from autistic researchers and advocates on non-ableist ways to elicit the views of autistic students, autistic participants were viewed from a strength-based lens with presumed competence [30] and, therefore, if they met the study's eligibility criteria, were considered to possess decision-making capacity to provide consent; therefore, parental consent was not required. However, participants and parents were provided information booklets describing the research, eligibility, participation requirements, participation risks, benefits, and contact information for mental health support should they require it. The booklet was carefully formatted to convey study information in a simple, concrete manner, paired with visual supports [31,32].

Eligible participants were between the ages of 15 and 21 and had finished at least one year of secondary school in Canada. In total, 72 surveys were included in the final data analysis, where most participants (80%, $n = 57$) had completed grade 10 and grade 11, and 20% ($n = 15$) of participants indicated that they had completed grade 12. Most participants self-identified as being diagnosed with Autism Spectrum Condition (74%, $n = 53$), Asperger's Syndrome (28%, $n = 15$), or Pervasive Developmental Disorder (6%, $n = 4$). Participants also indicated if they preferred speaking or non-speaking modes of communication, where approximately half of the participants (51%, $n = 37$) indicated that they preferred typing to communicate. Most participants shared that their primary motivation to participate in the research was the chance to receive a gift card ($n = 34/72$ or 47%), and nearly a quarter of participants ($n = 17/72$ or 24%) indicated that they received support to complete the survey. A summary of the participant demographics is provided in Table 1.

**Table 1.** Participant Demographics.

| Demographics | | *n* | % |
|---|---|---|---|
| Diagnostic Identity | | | |
| | ASC | 53 | 74% |
| | AS | 15 | 28% |
| | PDD | 4 | 6% |
| Gender Identity | | | |
| | Cis Female | 26 | 36% |
| | Cis Male | 40 | 56% |
| | Gender Queer | 1 | 1% |
| | Non-binary | 3 | 4% |
| Grade Completed | | | |
| | Gr. 10 | 40 | 56% |
| | Gr. 11 | 17 | 24% |
| | Gr. 12 | 15 | 20% |
| Communication Preference | | | |
| | Speaking | 31 | 43% |
| | Typing | 37 | 51% |
| | Other | 4 | 5% |
| Support Received | | | |
| | Yes | 17 | 24% |
| | No | 55 | 76% |

ASC = Autism Spectrum Condition, AS = Autism Spectrum, PDD = Pervasive Developmental Disorder.

### 2.2. Measures

The survey items in the original study were from the Panorama Survey [33], an open-source survey developed by the Harvard Graduate School of Education. The Panorama Survey consists of ten scales related to classroom climate, engagement, grit, learning strategies, mindset, pedagogical effectiveness, rigorous expectations, school belonging, teacher–student relationships, and valuing of the school. The original study included eighteen items selected from eight scales based on their likeliness to gather relevant student perception data to answer the research question.

### 2.3. Student Motivation

Student motivation was measured using the following six items from the Panorama Survey: (a) How comfortable are you asking questions about what you are learning in school; (b) How interesting is school; (c) How excited are you to go to school; (d) How eager are you to participate in school; (e) How many of your teachers would you be excited to see again in the future; and (f) Overall, how much do you feel you belong at your school. Responses were on a 4-point Likert scale (0 = almost never to 3 = almost always), except for the first item, which was on a 5-point Likert scale (0 = not to 4 = extremely). Responses to these items were summed to create a student motivation score. Cronbach's alpha for the six items was 0.91, indicating a high reliability.

### 2.4. Environmental, Teacher, and Peer Factors

Additional items from the Panorama Survey were selected to represent the school environment, teacher, and peer factors. These items were used in the final analyses. The following three items measured the environmental factors: (a) How fair or unfair are the rules at school; (b) How pleasant or unpleasant is the physical space in your school; and (c) How positive or negative is the energy at school? The environment items were on a 5-point Likert scale, with higher responses being more positive (0 = not to 4 = extremely). The following five items measured teacher factors: (a) How often do you receive feedback; (b) How often do your teachers take time to ensure you understand the material; (c) Overall, does your teacher have high expectations of you; (d) How many of your teachers are respectful towards you; and (e) How much do your teachers encourage you to do your best. Peer factors were measured by three items, which included: (a) How much respect do your peers show you; (b) How often do you worry about bullying at school; and (c) How well do your peers understand you as a person? Teacher and peer factors were on a 4-point Likert scale (0 = almost never to 3 = almost always).

### 2.5. Procedures

Often, people with disabilities are overlooked in research because ableist views on the legitimate forms of contribution and the representation of knowledge restrict participation [34]. Rigid recruitment procedures and limited access to accommodations help to explain the under-representation of first-person perspectives in disability-related research [35]. To promote accessibility in the study, recruitment procedures included emails sent to provincial and territory autism agencies and a post in MTurk. MTurk is a reliable crowdsource platform for accessing research subjects, including hard-to-reach [36] and people with disabilities [20,37]. The MTurk post closed after 50 surveys were completed, with two additional surveys counted because participants required additional time to complete. The ideal number of survey completions was determined by estimating that five participants from each province and territory would complete the survey.

To further strengthen the methodological design by reducing the participation barriers and promoting authentic neuro-divergent engagement from participants, recommendations from the Person-Oriented Autism Research Ethics [38] task force were followed, including (a) individualization; (b) an acknowledgment of the lived world; (c) respect for holistic personhood; (e) empowerment in decision-making; and (e) a focus on researcher–participant relationships. The practical implementation of these recommendations meant

that participants completed the survey online using Qualtrics, which aligns with research that states that autistic people prefer electronic communication over other forms [30,39], especially when interacting with unfamiliar people [40]. To support comprehension, survey items were paired with a simple graphic [30], participants could access familiar supports if required [41], and a completion bar was included at the bottom of each page to inform participants of their progress visually. Further benefits of the online survey are that participants did not have to navigate unfamiliar people and environments or rely on speaking forms of communication to participate.

### 2.6. Summary of Survey Completion

Seventy-five online surveys were completed; however, three surveys were removed from the final count because one participant completed the survey twice, and two entries were identified as spam by Qualtrics. Most participants resided in Ontario ($n = 23/72$ or 32%), British Columbia ($n = 20/72$ or 28%), and Alberta ($n = 15/72$ or 21%).

### 2.7. Data Analysis

A linear regression analysis was completed to investigate to what extent environmental, teacher, and peer factors were associated with autistic students' school motivation. The outcome variable was the student motivation composite. Each environmental, teacher, and peer item was included in the final model while controlling for gender, diagnostic identity, and the preferred communication method. Given that item-level data was used, we completed tests to see if the data met the assumption of collinearity. Results indicated that multi-collinearity was not a concern (tolerance = 0.22–0.88; *VIF* = 1.13–4.49).

## 3. Results

First, we examined the descriptive results for the motivation composite ($M = 10.05$, $SD = 5.28$; Range = 0–19). Second, we examined results from the regression, which indicated that the two unique predictors of autistic students' school motivation were (a) the environmental factor of how pleasant or unpleasant is the physical space in your school ($\beta = 0.93$, $p = 0.02$), and (b) the peer factor of how well your peers understand you as a person ($\beta = 2.95$, $p < 0.001$). This finding means that autistic students who rated their schools as having more pleasant physical spaces and that their peers understood them as a person had higher school motivation. In addition, students who indicated that their preferred communication method was speaking had higher motivation than students whose preferred communication method was typing ($\beta = 1.96$, $p = 0.03$). Teacher factors were not found to be related to autistic students' school motivation. A summary of this study's results is provided in Table 2.

**Table 2.** Factors Impacting Autistic Students' School Motivation.

| Construct | β | SE | p | Δr² |
|---|---|---|---|---|
| Constant | 3.34 | 1.71 | 0.06 | – |
| Gender | 0.49 | 0.79 | 0.54 | 0.10 |
| Asperger's | 1.27 | 0.91 | 0.17 | 0.80 |
| PDD | −1.59 | 1.19 | 0.19 | 0.80 |
| **Preferred Communication: Speaking** | **1.96** | **0.85** | **0.03** | **2.30** |
| School: Fair Rules | 0.75 | 0.42 | 0.08 | 1.40 |
| **School: Pleasant Physical Space** | **0.93** | **0.38** | **0.02** | **2.40** |
| School: Energy | −0.68 | 0.39 | 0.09 | 2.60 |
| Teacher: Feedback | −0.23 | 0.51 | 0.65 | 0.10 |
| Teacher: Ensure understanding | 0.55 | 0.53 | 0.31 | 0.50 |
| Teacher: High Expectations | 0.07 | 0.60 | 0.90 | 0.10 |
| Teacher: Respectful | −1.23 | 0.67 | 0.07 | 1.00 |
| Teacher: Encourage | 0.10 | 0.67 | 0.88 | 0.00 |
| Peer: Respect | 1.02 | 0.71 | 0.16 | 0.60 |
| Peer: Bullying | −0.79 | 0.49 | 0.12 | 8.30 |
| **Peer: Understanding** | **2.95** | **0.65** | **<0.001** | **1.90** |

## 4. Discussion

This study presents two main findings on the factors related to autistic secondary school students' motivation, which are: (a) students who rated their schools as having more pleasant physical spaces experienced higher levels of school motivation, and (b) students who felt understood by their peers also experienced higher levels of school motivation. These findings are discussed in turn.

### 4.1. Sensory-Friendly School Spaces and Student Motivation

The findings suggest that pleasant physical spaces were positively related to autistic students' school motivation. Qualitative results from the original mixed-method study found that the five main themes affecting autistic students' school experiences were as follows: teacher characteristics, peer interactions, sensory inputs from school environments, and communication factors [23]. Specifically, for the sensory inputs from school environments, the subthemes indicated that autistic students preferred sensory-friendly spaces (e.g., fluorescent lighting, obnoxious smells, and loud or busy spaces) [23]. These findings align with research that shows repeated exposure to undesirable and overwhelming sensory stimuli causes physical and emotional distress for many autistic students, leading to school avoidance and disengagement [28,42,43]. Autistic students also report that their concentration capability is significantly reduced in noisy, busy, and visually distracting environments [27], which, over time, may impact student achievement [44] and, therefore, motivation. Autistic post-secondary students also report that access to sensory-friendly campus spaces increases school engagement and achievement and contributes to their overall well-being [9,21,45]. To help to minimize the effects of physical and psychological stressors associated with overwhelming school spaces, neurodivergent students and researchers suggest providing flexible access to lectures, including in person, online, or hybrid options [46]. Moreover, a recent study investigating enabling and disabling environmental factors found that adaptations to the environment, enhancing predictability, and providing students with opportunities to recover from sensory overload were effective strategies [47]. These results suggest that more evidence and research on sensory-inclusive spaces for autistic people is needed, given the significant impact on mental health and the quality of life, including access to and participation in education and community [47].

### 4.2. Peer Understanding and School Motivation

A second factor related to autistic students' school motivation was their perceptions of peer understanding, which included an awareness of autism and an acceptance of diversity. Previously published qualitative data showed that when students described their positive school experiences, they used terms related to acceptance, understanding, and feeling supported by their peers [23]. Peer acceptance and an awareness of neurodiversity are reflective of school cultures committed to improving diversity, equity, and inclusion in education [46]. However, feeling understood by others depends on multiple experiences of meaningful social connections, which can be difficult for autistic people because of unusual interests and behaviors, communication barriers, and the misperceptions of social interests [48]. Further complicating instances of peer understanding are communication errors and social "breakdowns" that prevent autistic students from feeling equal to their peers [30]. However, autistic advocates have challenged neurotypical norms on appropriate social communication, suggesting that this issue is not one-sided but, instead, the consequence of mutual misunderstandings, which has been named the "double empathy problem" by autistic scholar Damian Milton [49]. Often, autistic students are expected to bridge social gaps by developing "appropriate" social skills, which over time can lead to self-rejection and perceptions that they need to mask parts of themselves in order to be accepted [50].

An unexpected finding in this study is that autistic students' school motivation was related to their communication preference; specifically, students who indicated that they preferred speaking over non-speaking modes (i.e., typing) had higher rates of school motivation. This finding might be explained by data that shows autistic students who

rely on non-speaking modes of communication, such as augmentative and alternative communication devices (AAC), face additional barriers to learning and school participation because of ableist assumptions that equate a speaking voice with an intelligent voice [51,52], and because staff are unsure on the effective ways to support minimally and non-speaking autistic students in school [53]. Hence, collaborating with autistic students to support their communication success should be a priority for stakeholders, especially given that up to one-third of the autistic population is minimally or non-speaking [54].

### 4.3. Implications

These preliminary findings on factors affecting autistic students' school motivation are significant, not only because they are based on the perspectives of autistic students, but also because they draw attention to the potential negative outcomes of over-looking school factors externally located from the student (i.e., the physical environment and social interactions) that impact autistic students' motivation, hence, school participation and success. While much research emphasizes a need for a greater professional understanding of autism [55], teacher preparation programs and professional development sessions, predominantly based on outsider perspectives, tend to perpetuate ableist biases and the use of pedagogical practices that hyper-focus on fixing traits internal to the autistic student, including behavioral-based interventions that attribute a lack of engagement to avoidance or attention-seeking behaviors. The risks associated with implementing support and strategies based on outsider perspectives are that they tend to be based on deficit views of ability [56] and are more likely to be abandoned or rejected by students [28], thus emphasizing the need for educators to engage in proactive and collaborative information-gathering discussions with autistic students to better understand their perspectives on the specific enablers and barriers affecting their motivation and engagement in learning and to promote a shared responsibility for finding positive solutions.

### 4.4. Future Directions

Addressing ableist biases and practices in research and education by incorporating the views of autistic students helps to ensure that (a) evidence-based practices are, in fact, responsive to the individual needs and contexts of students and (b) result in improved learning and well-being outcomes. The benefits of addressing environmental and social factors affecting autistic students' motivation extend beyond secondary school environments because they count for non-academic factors identified in research as negatively affecting autistic students' post-secondary enrollment and completion. Extending beyond the discussion of the relationship between student motivation and school success for both secondary and post-secondary autistic students is the potential and probable negative effects on autistic individuals' mental health, as they also experience higher rates of school failure and unemployment [14,16]. Findings from this study suggest that improving external factors affecting autistic students' school motivation, including improvements to school-based diversity, equity, inclusion, and accessibility initiatives, may improve short- and long-term outcomes, including those related to academic achievement and psychological well-being. Future studies could also examine effective strategies through the lens of factors impacting student motivation, rather than traditional deficit-based assessment practices.

### 4.5. Limitations

The secondary analysis conducted for this study highlights important and emergent findings. However, understanding the unique factors affecting autistic students' motivation based on self-reports would strengthen these preliminary findings and allow teachers and practitioners to implement responsive strategies in contextually relevant neuro-affirming ways. Also, while participants were encouraged to seek support to complete the survey, should they require aid (such as a reader to support with comprehension), the wording of some of the survey questions may have needed clarification for some students. Lastly, people from the autism community did not review the data or discuss the findings.

## 5. Conclusions

Preliminary findings suggest that for autistic students attending secondary school, sensory and social experiences affect their day-to-day experiences and, thus, their motivation to attend school and participate in social activities. Furthermore, for autistic students who do not use verbal speech as their dominant or preferred mode of communication, their school motivation is negatively impacted. These findings offer important considerations for educators and parents because they highlight the external factors affecting autistic students' school engagement and performance, therefore, highlighting the need for a more balanced and shared responsibility among stakeholders in creating enabling conditions for autistic students. Additionally, a knowledge of the factors affecting autistic students' school motivation can be used to improve the validity of investigative practices, such as academic and behavioral assessments, and consequently, the implementation of evidence-based interventions and the identification of supports to improve educational and well-being outcomes and graduation rates for autistic students.

**Author Contributions:** Conceptualization, C.L.; Methodology, C.L. and J.D.; Formal analysis, J.D.; Investigation, C.L.; Writing—original draft, C.L.; Writing—review & editing, J.D.; Project administration, C.L. version of the manuscript. All authors have read and agreed to the published version of the manuscript.

**Funding:** This research received no external funding.

**Institutional Review Board Statement:** The original study was approved by the University of Calgary's Conjoint Faculties Research Ethics Board (Ethics ID REB19-0185).

**Informed Consent Statement:** Informed consent was obtained from all subjects involved in the study.

**Data Availability Statement:** The data presented in this study are available on request from the corresponding author. The data are not publicly available due to privacy concerns.

**Acknowledgments:** The authors are very grateful to the participants who gave their time and shared their experiences with us by completing the survey. All authors certify that they have no affiliations with or involvement in any organization or entity with any financial or non-financial interest in the subject matter or materials discussed in this manuscript.

**Conflicts of Interest:** The authors declare no conflicts of interest.

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
