# Peer review of "Factors Affecting Autistic Students’ School Motivation"

_education, doi:10.3390/educsci14050527_

Round 1

Reviewer 1 Report

Comments and Suggestions for Authors

Overall, this is an interesting, well written and organised paper which reports on findings from online survey data.  However, I have a few concerns which I feel need further attention prior to consideration for publication.   

These are:

In the Introduction, data is provided detailing the percentage of young people who finish or do not complete various stages of secondary schooling in the USA. However, this research appears to be based in Canada.  The reader immediately becomes unclear about whether the issue of secondary school completion for young people with autism is the same in Canada as in USA.  This needs to be clarified especially as the journal’s readership is international and further contextual information would be beneficial. 

I am concerned about the lack of detail written about ethics.  I understand that the researchers were informed by the Person -Oriented Autism Research Ethics, but did they gain permission for the research from an institution and did they seek to gain other forms of consent from the participants and their guardians (for the younger participants).  If, for example, the participants self-identified their neurodivergence how assured were the researchers that this was accurate reporting?  Also, did they put into place any support post survey for any emotional trauma caused to the participants by answering / engaging with the research.  The ethical process and safeguards need to be more thorough and transparent.

In line with the above comments, further details about the analysis process would add strength to the paper.

Finally, I feel that the lack of inclusion of the qualitative data limited the validity and robustness of some of the conclusions proposed.  For example, stating that the participants were most likely to be referring to ‘more sensory – friendly’ environments when the word ‘pleasant’ was used in the survey (p.6, section 4.1) was not clearly supported by any evidence detailed in the paper.  Using ‘inferred’ immediately suggests that the researchers placed more reliance on information from previous research literature than on the actual evidence in their study.  I suspect that the qualitative data gathered as part of the larger study has evidence which would add further support to these conclusions, but without this information the reader might well question the reliability of the paper’s conclusions. 

I hope the author (s) find these comments constructive and helpful.

Reviewer 2 Report

Comments and Suggestions for Authors

The questions posed to the students: I wonder if all students with SEN would understand questions such as: How pleasant or unpleasant is the physical space in your school, and 127 (c) How positive or negative is the energy at school? Were additional explanations provided? I think this needs to be clarified.

It would be best to define motivation as this is something that can have multiple definitions. A short discussion on motivation and clearly stating the definition being used in the study would be helpful.

Future studies could also examine how school assess the needs to autistic students. For example, sensory differences are now a core part of Autism according to the DSM V but these differences are individual. For this reason, an assessment of need is necessary to enable schools to individualise supports for students. This would also assist schools to put in place proactive strategies to support students.

I agree that qualitative data, particularly gained from students, would strengthen this study.

The questions posed to the students: I wonder if all students with SEN would understand questions such as: How pleasant or unpleasant is the physical space in your school, and 127 (c) How positive or negative is the energy at school? Were additional explanations provided? I think this needs to be clarified.

It would be best to define motivation as this is something that can have multiple definitions. A short discussion on motivation and clearly stating the definition being used in the study would be helpful.

Future studies could also examine how school assess the needs to autistic students. For example, sensory differences are now a core part of Autism according to the DSM V but these differences are individual. For this reason, an assessment of need is necessary to enable schools to individualise supports for students. This would also assist schools to put in place proactive strategies to support students.

I agree that qualitative data, particularly gained from students, would strengthen this study.
